# Vesicular Zinc Modulates Cell Proliferation and Survival in the Developing Hippocampus

**DOI:** 10.3390/cells12060880

**Published:** 2023-03-11

**Authors:** Selena Fu, Ashley T. Cho, Simon C. Spanswick, Richard H. Dyck

**Affiliations:** 1Hotchkiss Brain Institute, University of Calgary, Calgary, AB T2N 1N4, Canada; selena.fu2@ucalgary.ca (S.F.);; 2Department of Psychology, University of Calgary, Calgary, AB T2N 1N4, Canada; 3Alberta Children’s Hospital Research Institute, University of Calgary, Calgary, AB T2N 1N4, Canada; 4Department of Cell Biology and Anatomy, University of Calgary, Calgary, AB T2N 1N4, Canada

**Keywords:** development, neurogenesis, hippocampus, cell proliferation, cell survival, zinc, zinc transporter 3

## Abstract

In the brain, vesicular zinc, which refers to a subset of zinc that is sequestered into synaptic vesicles by zinc transporter 3 (ZnT3), has extensive effects on neuronal signalling and modulation. Vesicular zinc-focused research has mainly been directed to its role in the hippocampus, particularly in adult neurogenesis. However, whether vesicular zinc is involved in modulating neurogenesis during the early postnatal period has been less studied. As a first step to understanding this, we used ZnT3 knockout (KO) mice, which lack ZnT3 and, thus, vesicular zinc, to evaluate cell proliferation at three different age points spanning postnatal development (P6, P14, and P28). The survival and the neuronal phenotype of these cells was also assessed in adulthood. We found that male ZnT3 KO mice exhibited lower rates of cell proliferation at P14, but a greater number of these cells survived to adulthood. Additionally, significantly more cells labelled on P6 survived to adulthood in male and female ZnT3 KO mice. We also found sex-dependent differences, whereby male mice showed higher levels of cell proliferation at P28, as well as higher levels of cell survival for P14-labelled cells, compared to female mice. However, female mice showed greater percentages of neuronal differentiation for P14-labelled cells. Finally, we found significant effects of age of BrdU injections on cell proliferation, survival, and neuronal differentiation. Collectively, our results suggest that the loss of vesicular zinc affects normal proliferation and survival of cells born at different age points during postnatal development and highlight prominent sex- and age-dependent differences. Our findings provide the foundation for future studies to further probe the role of vesicular zinc in the modulation of developmental neurogenesis.

## 1. Introduction

Zinc is an essential trace element that serves important structural, enzymatic, and regulatory functions in the tissues and cells of the mammalian body [1,2,3,4]. As the second-most abundant transition metal after iron, its prevalence is a reflection of its widespread functions, which include regulating cell proliferation and differentiation [1,5,6,7], cell apoptosis [8,9], gene expression and transcription factors [10,11,12], and stabilizing enzyme structures [13]. In addition to its role in basic cellular functioning, the maintenance of zinc homeostasis in the central nervous system also proves vital, the dysregulation of which has been implicated in a wide array of neurological disorders, including neurodevelopmental disorders such as autism spectrum disorder [14,15,16,17], mood disorders such as depression and anxiety [18,19], and neurodegenerative diseases such as Alzheimer’s and Parkinson’s disease [20,21,22].

In the brain, the majority of zinc is bound to proteins. However, there exists a pool of unbound “free” zinc that is sequestered into presynaptic vesicles of a select subset of glutamatergic neurons [1,4,23,24,25]. This localized pool of free zinc is referred to as “vesicular zinc” and constitutes approximately 5–15% of the total zinc in the brain [23,25,26]. Once packaged, vesicular zinc is co-released with glutamate into the synaptic cleft in a voltage- and calcium-dependent manner, where it can exert signalling effects by binding to and acting upon several postsynaptic receptors [27]. The sequestration of zinc ions into synaptic vesicles is inextricably dependent on vesicular zinc transporter 3 (ZnT3), encoded by the *SLC30A3* gene [23,28,29]. Without ZnT3, as in the case of the genetically modified ZnT3 knockout (KO) mouse, vesicular zinc is undetectable in the brain [30].

The vesicular zinc levels and ZnT3 expression are especially highest in the hippocampus, a region involved in learning and memory. The hippocampus is also notable for harbouring one of only two regions of the brain that continues to produce new neurons throughout life, a process referred to as neurogenesis [31,32,33]. The abundance of vesicular zinc in the hippocampus suggests it may play a role in hippocampal function and neurogenesis, and indeed, previous studies have revealed that adult ZnT3 KO mice exhibit subtle cognitive deficits in certain learning and memory behavioural domains [34,35]. With respect to adult hippocampal neurogenesis, ZnT3 KO mice have been found to have reduced neuroblast production in the dentate gyrus [36]. Interestingly, in circumstances where neurogenesis is upregulated, such as in response to hypoglycaemia, ZnT3 KO mice display significantly fewer proliferating progenitor cells and immature neurons [37]. Previous unpublished experiments from our laboratory provide further support for this. Normally, neurogenesis, cell survival, and learning and memory abilities, including that of spatial memory in rodents, are strongly enhanced when housed in an enriched environment consisting of a larger living space, a greater number of mice living together, climbing ladders, running wheels, tunnels, and a variety of toys [38,39,40,41,42]. However, in our previous experiments, ZnT3 KO mice did not show an increase in neurogenesis nor an improvement of spatial memory following environmental enrichment. Moreover, ZnT3 KO mice had significantly decreased levels of cell survival, demonstrating the necessity of vesicular zinc for increased hippocampal neurogenesis, sustained cell survival, and behavioural benefits induced by environmental enrichment [43].

The role of zinc in developmental neurogenesis has also been well described, though most of the previous research has utilised a maternal dietary zinc deficiency. Depriving suckling rat pups of zinc by feeding nursing dams a zinc-deficient diet was found to result in impaired body growth; alterations in the composition of the hippocampus and cerebellum; and smaller forebrains, cerebella, and hippocampi in the rat pups [44,45]. At the cellular level, gestational zinc deficiency has been found to significantly reduce the proliferation of neural precursor cells, which was associated with an overall decrease in the total number of neurons in young adulthood [46,47], while postnatal zinc deficiency has been found to reduce the differentiation of basket and stellate cell dendrites, delay the maturation of Purkinje cells, and decrease the number of granule cell neurons in the cerebellar cortex [48,49,50]. In addition to the structural and biochemical abnormalities, zinc deprivation during the foetal, lactation, and early postnatal periods has been found to induce several irreversible deficits in learning and memory that persist into adulthood [51,52,53]. For instance, rats exposed to zinc deficiency during lactation and early infancy display impairments in long-term and working memory in adulthood [51,52]. In another study, male offspring of dams deprived of zinc during the latter third of pregnancy display an apparent increase in sensitivity to shock in an avoidance condition task, which resulted in a decline in learning [54]. Interestingly, females did not display any impairments in active avoidance, suggesting potential sex differences in response to intrauterine zinc deprivation [54].

While these studies demonstrate the consequences of zinc deprivation during early development on hippocampal neurogenesis and learning and memory, utilizing a systemic manipulation such as a dietary zinc deficiency model does not differentiate between the role of zinc as an essential mineral within normative cellular function and vesicular zinc. Given the lack of specificity attained, the extent to which these methods lead to alterations in vesicular zinc levels and whether the changes are directly attributable to a reduction in vesicular zinc are circumstantial. Furthermore, the existing neurogenesis studies that do examine the specific role of vesicular zinc by use of the ZnT3 KO mouse model have been dedicated to studying adult neurogenesis, not early postnatal neurogenesis. As a result, data regarding the specific role of vesicular zinc in modulating neurogenesis in the developing hippocampus remains unclear.

As a first step to understanding whether vesicular zinc is involved in modulating hippocampal neurogenesis in the developing mouse brain, we examined cell proliferation in the dentate gyrus of male and female ZnT3 KO and wildtype (WT) mice at three developmental age points (postnatal day (P) 6, 14, and 28). Additionally, cell survival and neuronal differentiation were assessed in adulthood (P60) to determine whether the phenotype and survival of cells born at each developmental age point were affected by the absence of vesicular zinc. We hypothesized that the lack of vesicular zinc would differentially affect cell proliferation, survival, and differentiation, relative to WT mice. As sex differences have been suggested to occur in the ZnT3 KO mice [55] and in the regulation of neurogenesis [56,57], we also sought to explore whether there were any differences between male and female mice. Finally, given that rates of neurogenesis and neuronal survival vary across stages of the lifespan [58], we explored whether there were any differences in the proliferation, survival, and differentiation across the three developmental age points during which cells were labelled. We hypothesized there would be differences between male and female mice, as well as differences in cell proliferation, cell survival, and neuronal differentiation as a function of age.

## 2. Materials and Methods

### 2.1. Animals

All experimental procedures were approved by the Life and Environmental Sciences Animal Care Committee at the University of Calgary (Study ID: #AC20-0141) and were performed in accordance with the guidelines regarding the humane and ethical treatment of animals set by the Canadian Council on Animal Care. The WT and ZnT3 KO mice for this study were from a mixed C57BL/6 × 129/Sv genetic background and bred from homozygous breeding pairs in the Life and Environmental Science Animal Research Centre vivarium at the University of Calgary.

A total of 144 mice consisting of 72 male (WT: *n* = 36 and ZnT3 KO: *n* = 36) and 72 female (WT: *n* = 36 and ZnT3 KO: *n* = 36) mice were used for this experiment. Offspring were housed with both parents until P21; at which point, they were weaned and then group-housed (2–5 per cage) with same sex littermates. All mice were living in clear, polycarbonate standard housing cages (28 × 17.5 × 12 cm) lined with woodchip bedding, sizzle paper, nesting material, and one enrichment object. Cages were kept in temperature- and humidity-controlled rooms (22 °C; 25–30% humidity) on a 12:12 h light/dark cycle (lights on during the day). Food and water were provided ad libitum. To birthdate animals, pregnant dams were checked daily for litters. P0 was defined as the first day that pups were observed in the cage by experimenters. Litters ranged in size from 2–14 pups.

### 2.2. Experimental Design

For a diagram depicting the experimental design and timeline, see Figure 1. Three different age points (P6, P14, and P28) were used to examine cell proliferation, and P60 was used to examine cell survival. Selection of the age points to examine cell proliferation was based upon the ages at which ZnT3 and vesicular zinc expression are detectable [59,60]. Additionally, by selecting two younger ages (P6 and P14) and one relatively more mature age (P28), a bigger window of development could be examined. Male and female WT and ZnT3 KO pups from each litter were assigned to one of the three postnatal age groups (P6, P14, or P28); at which point, they received two subcutaneous injections (50 mg/kg) of the thymidine analogue 5′-bromo-2′-deoxyuridine (BrdU; B5002, Sigma-Aldrich, Oakville, ON, Canada) dissolved in sterile 0.9% saline at approximately 09:00 and 11:00 h. Within each of these three postnatal age groups, half of the pups were killed approximately 24 h after the last BrdU injection to assess cell proliferation. Thus, tissue was harvested from 12 groups to assess cell proliferation: mice that received BrdU at P6 (male WT: *n* = 6; male ZnT3 KO: *n* = 6; female WT: *n* = 6; female ZnT3 KO: *n* = 6), P14 (male WT: *n* = 6; male ZnT3 KO: *n* = 6; female WT: *n* = 6; female ZnT3 KO: *n* = 6), and P28 (male WT: *n* = 6; male ZnT3 KO: *n* = 6; female WT: *n* = 6; female ZnT3 KO: *n* = 6). The remaining pups were housed with both parents until they were weaned at P21 and then group-housed (2–5 per cage) with same sex littermates until P60; at which point, they were killed to assess cell survival. Thus, tissue was harvested from 12 groups to assess cell survival at P60: mice that received BrdU at P6 (male WT: *n* = 6; male ZnT3 KO: *n* = 6; female WT: *n* = 6; female ZnT3 KO: *n* = 6), P14 (male WT: *n* = 6; male ZnT3 KO: *n* = 6; female WT: *n* = 6; female ZnT3 KO: *n* = 6), and P28 (male WT: *n* = 6; male ZnT3 KO: *n* = 6; female WT: *n* = 6; female ZnT3 KO: *n* = 6).

To mitigate the litter effects, male and female pups from each litter were distributed as equally as possible across the proliferation and survival groups (i.e., pups that were killed either 24 h after their last BrdU injection or left to survive to P60). In addition, experimental groups consisted of mice that were selected from different litters and breeding pairs whenever possible. We endeavoured to ensure that any offspring of the same breeding pair (but in different litters) were not assigned to the same postnatal age group as the first litter. All these considerations were contingent upon the remaining number of mice needed for each experimental group.

### 2.3. Tissue Preparation

Mice were injected intraperitoneally with an overdose of Euthanyl (240 mg/mL sodium pentobarbital; Bimeda, Cambridge, ON, Canada) and transcardially perfused with 0.1M phosphate-buffered saline (PBS) containing heparin (54 mg/L; H4784, Sigma-Aldrich, Oakville, ON, Canada) until the blood was cleared, followed by 4% paraformaldehyde (PFA) in PBS. Brains were extracted and post-fixed by immersion in 4% PFA in 0.1M PBS at 4 °C for three days. After post-fixing, the brains were transferred to a sucrose solution (30% sucrose and 0.02% sodium azide in 0.1M PBS) and stored at 4 °C until they were no longer buoyant. Brains were hemisected by a midsagittal cut through the longitudinal fissure to produce the left and right hemispheres. Each hemisphere was examined for any gross anatomical defects or tissue damage, and the hemisphere with the least or no deformities was selected for sectioning. In cases where both hemispheres appeared intact with no damage, one hemisphere was arbitrarily selected, as we did not hypothesise any differences between the hemispheres. The selected hemisphere was cut into six series of 40 µm sections in the sagittal plane using a freezing, sliding microtome (American Optical, Model #860; Buffalo, NY, USA). Sections were then stored in 0.02% sodium azide in 0.1M PBS until immunohistochemical processing.

The choice of orientation for the brain sections was based upon several considerations. Sagittal sections provide the future opportunity to simultaneously examine the subventricular zone (SVZ), rostral migratory stream, and olfactory bulb. The SVZ is the only other region of the brain where adult neurogenesis occurs, and new neurons from the SVZ migrate to the olfactory bulb via the rostral migratory stream [61,62]. Additionally, it was more time-efficient, since only one hemisphere was needed for sectioning.

### 2.4. BrdU Immunohistochemistry

To detect BrdU-positive (BrdU^+^) cells, a one-in-three series of brain tissue sections was stained using diaminobenzidine (DAB) immunohistochemistry. Prior to immunohistochemical processing, antigen retrieval was performed by incubating the sections in 2N hydrochloric (HCl) acid for 30 min at 45 °C [63]. The HCl treatment was neutralised by rinsing sections in 0.1 M sodium borate buffer (pH 8.5) for 10 min, followed by 3 × 10 min washes in 0.1M PBS. Sections were then incubated with purified mouse monoclonal (Bu20a) anti-BrdU primary antibody (1:1000, BioLegend Cat. No. 339802, San Diego, CA, USA) in blocking solution (3% normal horse serum and 0.03% Triton-X in 0.1M PBS) for 48 h at 4 °C. Thereafter, sections were rinsed three times for 10 min each in 0.1M PBS and subsequently incubated overnight at 4 °C with biotinylated horse anti-mouse secondary antibody (1:1000, Vector Laboratories Cat. No. BA-2000-1.5, Burlingame, CA, USA) in 0.03% Triton-X in 0.1M PBS. Next, 3 × 10 min washes in 0.1M PBS preceded a one-hour incubation in an Avidin–Biotin Complex (VECTASTAIN Elite ABC-Peroxidase Kit, Mouse IgG, Cat. No. PK-6102, Vector Laboratories, Burlingame, CA, USA) at room temperature. Sections were then washed in 0.1M PBS twice, then once in 0.05M Tris buffer (pH 7.6) for 10 min each. Next, BrdU+ cells were visualized using Ni-DAB staining (1 mg/mL DAB in 0.05M Tris-HCl buffer (pH 7.6), with 1% nickel ammonium sulphate and 5% 1M imidazole). Immediately before use, 1% H2O2 was added for a final concentration of 0.03%. Sections were left to react in the Ni-DAB solution for approximately 13 min. Sections were then washed three times in 0.03% Triton-X in 0.1M PBS for 10 min each at room temperature before being serially mounted onto gelatine-coated slides. Sections were left to dry overnight before being counterstained with neutral red for 2.5 min. Excess stain was removed by briefly dipping the sections in distilled water before undergoing dehydration in an ascending ethanol series: 70% EtOH: 1 × 2 min, 95% EtOH: 1 × 2 min, and 100% EtOH: 2 × 1 min. Sections were then cleared using xylene (1 × 3 min) and transferred to fresh xylene prior to being coverslipped with Permount^TM^ Mounting Medium (Fisher Scientific). Slides were stored flat and allowed to dry prior to analysis.

### 2.5. Double-Label Immunofluorescence

Analyses of neuronal phenotypes were performed with fluorescent detection on an additional series of free-floating tissue sections taken from a subset of P60 mice that received BrdU injections on P6, P14, and P28 (*n* = 36; 3 mice per group). Prior to immunofluorescent processing, tissue sections were denatured in 2N HCl acid for 60 min at room temperature with gentle agitation every 15 min. Sections then underwent 6 × 15 min washes in 0.1M PBS [64]. Sections were then transferred to and incubated in a primary antibody solution containing rat anti-BrdU (1:500, Bio-Rad Laboratories Product Code MCA6144, Mississauga, ON, Canada), mouse anti-NeuN (1:1000, Millipore Sigma-Aldrich Product No. MAB377, Cat. No. 636574, Oakville, ON, Canada), 0.3% Triton-X, and 2% normal goat serum in 0.1M PBS for 24 h at 4 °C. Sections were then rinsed three times for 10 min each in 0.1M PBS and subsequently incubated in a secondary solution consisting of goat anti-rat biotinylated IgG (1:500, Jackson ImmunoResearch Laboratories Code No. 112-065-167, West Grove, PA, USA) and goat anti-mouse Alexa Fluor-488 (1:250, Jackson ImmunoResearch Laboratories Code No. 115-545-003) in 0.1M PBS for 24 h at 4 °C. Following 3 × 10 min washes in 0.1M PBS, tissue sections were then incubated in the tertiary solution containing Streptavidin Alexa Fluor-568 conjugate (1:500, Invitrogen Thermo Fisher Scientific Cat. No. S11226, Eugene, OR, USA) in 0.1M PBS for 60 min at room temperature. Following 3 × 10 min washes in 0.1M PBS, sections were serially mounted onto gelatine-coated slides before being coverslipped with Fluoromount-G ^TM^ Mounting Medium with DAPI (Invitrogen Thermo Fischer Scientific Cat. No. 00-4959-52). Slides were stored flat and in the dark at 4 °C until analysis.

### 2.6. Microscopy and Cell Quantification

The number of DAB-stained BrdU^+^ cells was counted in all sections spanning the entire dentate gyrus using a brightfield microscope (Zeiss Axioskop 2) with a 40×/0.75 objective. All counting was conducted by experimenters who were blinded to the sex and genotype of the animal to ensure unbiased quantification. To avoid overestimation errors, counting was performed by focusing through the thickness of the section, and cells in the uppermost and lowermost focal planes were excluded [63,65].

Two criteria were established for the proper identification and quantification of BrdU^+^ cells. The first criterion was morphology. Only uniformly labelled nuclei that were grey or black in colour, with a clearly defined membrane, were considered BrdU^+^. Cells that either lacked a distinct border or exhibited punctate staining in part of the nucleus were excluded from the study [66]. The second criterion for a BrdU^+^ profile was the location of the cells. In the P6 proliferation groups, the dentate gyrus was not yet fully formed, and many BrdU^+^ cells were found scattered throughout the hilus, granule cell layer (GCL), and molecular layer (Figure 2A,B) [67,68,69]. In contrast, in the P14 and P28 proliferation groups, BrdU^+^ cells were found almost exclusively within the GCL and subgranular zone (SGZ), a region defined as three cell widths from the hilar edge of the GCL (Figure 2E,F,I,J) [64]. As such, for the P6 proliferation groups, BrdU^+^ cells were quantified in both the hilus and SGZ/GCL to ensure that BrdU^+^ cells contributing to the formation of the dentate gyrus GCL were not unnecessarily excluded. Counts obtained from the hilus and SGZ/GCL were recorded separately, and figures depict the breakdown of the two; however, the aggregate was used for the statistical analysis. For the P14 and P28 proliferation groups and all survival groups, the BrdU^+^ cell counts were limited to the SGZ and GCL in all sections containing the hippocampal dentate gyrus (Figure 2C–L). All counts were multiplied by six to estimate the total number of BrdU^+^ cells.

To determine the phenotype of surviving BrdU^+^ cells, fluorescent signals were detected using an Olympus FluoView FV3000 confocal laser scanning microscope equipped with FV31S-SW viewer software and a UPlanXApo 60×/1.42 oil immersion objective. Co-localisation of BrdU and NeuN was assessed in ~25 randomly selected BrdU^+^ cells in the dentate gyrus of each mouse, and the percentage of BrdU^+^ cells co-labelled with NeuN was determined. BrdU/NeuN co-expression was confirmed using stacks of images (512 × 512 pixels) that were taken at a step size of 0.51 µm spanning the *z*-axis of the cell [70]. Z-stacks were viewed in the orthogonal plane to ensure there were no overlapping labelled cells in the depth of the stack, and a cell was considered positive for BrdU/NeuN if a nuclear signal from both the 488 nm and 561 nm lasers co-localised in the *x*-, *y*-, and *z*-axes [70,71].

### 2.7. Statistical Analysis

All statistical analyses were conducted using IBM SPSS Statistics (Version 26; Armonk, NY, USA: IBM^®^). Unless otherwise stated, comparisons were conducted by a three-way factorial analysis of variance (ANOVA) with sex (Male vs. Female), genotype (WT vs. ZnT3 KO), and age (P6 vs. P14 vs. P28) as the factors for each stage (Proliferation and Survival). Significant interactions were followed up with either a priori planned contrasts or post hoc Tukey’s Honestly Significant Difference multiple comparisons. All ANOVA results are reported in Appendix A. Tests for the assumption of normality (Shapiro–Wilk test of normality) and homogeneity of variances (Levene’s test for equality of variances) are reported in Appendix A. Data are reported as the mean ± standard deviation (Appendix A), unless otherwise stated. Figures were created using GraphPad Prism Software (Version 9; San Diego, CA, USA).

## 3. Results

### 3.1. Cell Proliferation in Mice Injected on P6, P14, and P28

To examine the cell proliferation between genotypes across the three developmental age points, as well as potential sex differences, the number of BrdU^+^ cells was examined for male and female WT and ZnT3 KO mice that received BrdU injections on P6, P14, and P28 and were killed approximately 24 h after the last injection. There was no three-way interaction between sex, genotype, and age [*F*(2, 60) = 1.47, *p* = 0.237], nor was there a two-way interaction between sex and genotype [*F*(1, 60) = 0.99, *p* = 0.323]. There was, however, a difference in the number of BrdU^+^ cells between WT and ZnT3 KO mice, depending on the age point examined [genotype × age interaction: *F*(2, 60) = 6.03, *p* = 0.004], as well as a difference between male and female mice also dependent on the age point examined [sex × age interaction: *F*(2, 60) = 5.95, *p* = 0.004]. There was also a statistically significant main effect of age [*F*(2, 60) = 4.71, *p* = 0.013] but no main effect of the genotype [*F*(1, 60) = 0.19, *p* = 0.664] or sex [*F*(1, 60) = 0.26, *p* = 0.609] (Appendix A).

To follow up the significant interaction between genotype and age, we conducted planned contrasts to test our a priori hypotheses that cell proliferation at the three developmental age points would differ between WT and ZnT3 KO mice. The results revealed there was no difference between genotypes in the number of BrdU^+^ cells of mice injected on P6 [*F*(1, 60) = 3.80, *p* = 0.056] or P28 [*F*(1, 60) = 0.04, *p* = 0.844]. There was a difference between genotypes of mice injected on P14, with WT mice displaying a greater number of BrdU^+^ cells compared to ZnT3 KO mice [*F*(1, 60) = 8.42, *p* = 0.005]. When this comparison was broken down by sex, we found that this was specific to male mice [*F*(1, 60) = 11.66, *p* = 0.001] (Figure 3). These results suggest that there were differences between genotypes in cell proliferation at P14 in male mice, with less proliferating cells seen in male ZnT3 KO mice compared to male WT mice.

Given we were also interested in sex differences, a post hoc simple effects test comparing male and female mice at each developmental age point was conducted to examine the significant interactions between sex and age. The results revealed there were no differences in the number of BrdU^+^ cells between male and female mice injected on P6 [*F*(1, 60) = 0.19, *p* = 0.668] or P14 [*F*(1, 60) = 2.87, *p* = 0.095]. There was, however, a difference between male and female mice injected on P28, whereby male mice had significantly higher numbers of BrdU^+^ cells compared to female mice [*F*(1, 60) = 9.10, *p* = 0.004] (Figure 3). These results indicate that, regardless of the genotype of the mouse, the sex influenced cell proliferation at P28, with male mice displaying greater rates of cell proliferation than female mice.

### 3.2. Age Effects on Cell Proliferation in Mice Injected on P6, P14, and P28

Given that the rate of neurogenesis changes across multiple stages of the lifespan [58,72], we were also interested in examining whether there were any effects of the age at which BrdU was administered on cell proliferation. To do this, we conducted a two-way ANOVA to compare the number of BrdU^+^ cells between WT and ZnT3 KO mice injected on P6, P14, and P28. A separate ANOVA was run for each sex (male and female) given the previous sex difference found in the cell proliferation (Appendix A).

Our results for male mice revealed there was no significant main effect of genotype [*F*(1, 30) = 1.11, *p* = 0.301] or age [*F*(2, 30) = 2.62, *p* = 0.089]. However, we found that the age at which BrdU was administered had differing effects on the number of BrdU^+^ cells, depending on the genotype of the mouse [genotype × age interaction: *F*(2, 30) = 7.26, *p* = 0.003]. Specifically, in male WT mice, there were no differences in the number of BrdU^+^ cells between mice injected on P6, P14, and P28 [*F*(2, 30) = 0.63, *p* = 0.541]. Conversely, there were differences in the number of BrdU^+^ cells across the developmental age points in male ZnT3 KO mice [*F*(2, 30) = 9.25, *p* < 0.001], with significantly less BrdU^+^ cells observed in mice injected on P14 compared to mice injected on P6 [*t*(30) = −4.17, *p* < 0.001] and P28 [*t*(30) = −3.00, *p* = 0.015]. There was no difference in cell proliferation between male ZnT3 KO mice injected on P6 and P28 [*t*(30) = 1.17, *p* = 0.479] (Figure 4A). These results indicate that the cell proliferation rates in male WT mice did not differ across the three developmental age points, whereas male ZnT3 KO mice had the lowest rate of cell proliferation at P14.

For female mice, there was a significant effect of age on the number of BrdU^+^ cells [*F*(2, 30) = 7.66, *p* = 0.002], revealing that female WT and ZnT3 KO mice injected on P6 and P14 had significantly greater numbers of BrdU^+^ cells compared to the mice injected on P28 [P6 vs. P28: *t*(30) = 3.71, *p* = 0.002; P14 vs. P28: *t*(30) = 2.93, *p* = 0.017]. There was no difference in the number of BrdU^+^ cells between female mice injected on P6 and P14 [*t*(30) = 0.78, *p* = 0.716] (Figure 4B). There was no significant genotype [*F*(1, 30) = 0.15, *p* = 0.705] or genotype × age interaction [*F*(2, 30) = 0.73, *p* = 0.489] effect on the number of BrdU^+^ cells. Altogether, these results indicate that female WT and ZnT3 KO mice had greater rates of cell proliferation during the first and second weeks of postnatal development.

### 3.3. Cell Survival in Mice Injected on P6, P14, and P28

Next, we assessed cell survival between genotypes, as well as potential sex differences, by examining the number of BrdU^+^ cells that remained at P60 for male and female WT and ZnT3 KO mice that received BrdU injections on P6, P14, and P28. There was no three-way interaction between sex, genotype, and age [*F*(2, 60) = 2.25, *p* = 0.114], nor was there a two-way interaction between sex and genotype [*F*(1, 60) = 0.09, *p* = 0.767]. There was, however, a difference in the number of BrdU^+^ cells between WT and ZnT3 KO mice, depending on the developmental age point examined [genotype × age interaction: *F*(2, 60) = 9.41, *p* < 0.001], as well as a difference between male and female mice also dependent on the developmental age point examined [sex × age interaction: *F*(2, 60) = 4.24, *p* = 0.019]. There was also a statistically significant main effect of age [*F*(2, 60) = 130.10, *p* < 0.001] and genotype [*F*(1, 60) = 27.34, *p* < 0.001] but no main effect of sex [*F*(1, 60) = 2.44, *p* = 0.124] (Appendix A).

To follow up the significant interaction between genotype and age, we conducted planned contrasts to test our a priori hypotheses that the number of BrdU^+^ cells remaining at P60 would differ between WT and ZnT3 KO mice injected on the three different developmental age points. The results revealed that ZnT3 KO mice injected on P6 and P14 had a greater number of BrdU^+^ cells that survived to P60 compared to WT mice [P6: *F*(1, 60) = 36.53, *p* < 0.001; P14: *F*(1, 60) = 9.61, *p* = 0.003]. When these comparisons were broken down by sex, we found that both male [*F*(1, 60) = 12.78, *p* < 0.001] and female [*F*(1, 60) = 24.72, *p* < 0.001] ZnT3 KO mice injected on P6 had significantly greater levels of BrdU^+^ cell survival at P60. In contrast, while both male and female ZnT3 KO mice injected on P14 displayed higher levels of remaining cells at P60 compared to WT mice, this effect was only statistically significant in male mice, with 45% more BrdU^+^ cells observed at P60 in male ZnT3 KO mice than male WT mice [*F*(1, 60) = 12.34, *p* < 0.001] (Figure 5). There were no differences in the number of BrdU^+^ cells that survived when assessed at P60 between genotypes of mice injected on P28 [*F*(1, 60) = 0.01, *p* = 0.930]. Overall, these results suggest there were differences between genotypes in cell survival, with greater levels of BrdU^+^ cells remaining at P60 in male and female ZnT3 KO mice injected on P6, as well as male ZnT3 KO mice injected on P14.

Given we were also interested in sex differences, a post hoc simple effects test comparing male and female mice at each developmental age point was conducted to examine the significant interactions between sex and age. The results revealed there were no differences in the number of BrdU^+^ cells remaining at P60 between male and female mice injected on P6 [*F*(1, 60) = 0.34, *p* = 0.561] or P28 [*F*(1, 60) < 0.01, *p* = 0.970]. There was, however, a difference between male and female mice injected on P14, whereby male mice had significantly higher numbers of BrdU^+^ cells remaining at P60 compared to female mice [*F*(1, 60) = 10.57, *p* = 0.002] (Figure 5). These results indicate that, regardless of the genotype of the mouse, the sex influenced cell survival in mice injected on P14, with greater rates of cell survival seen in male mice than female mice.

### 3.4. Age Effects on Cell Survival in Mice Injected on P6, P14, and P28

A series of two-way ANOVAs with the genotype (WT vs. ZnT3 KO) and developmental age point (P6 vs. P14 vs. P28) as factors was also conducted to examine whether there were any effects of the age at which BrdU was administered on cell survival, as measured by BrdU^+^ cells at P60. A separate ANOVA was run for each sex (male and female) given the previous sex differences found in cell survival (Appendix A).

For male mice, we found a significant main effect of the genotype [*F*(1, 30) = 14.20, *p* < 0.001] and age [*F*(2, 30) = 54.30, *p* < 0.001]. Additionally, there was also a significant interaction between the genotype and age [*F*(2, 30) = 4.63, *p* = 0.018], indicating that the age at which BrdU injections was administered had differing effects on the number of BrdU^+^ cells that remained at P60, depending on the genotype of the mouse. Specifically, male WT mice injected on P6 had a significantly greater number of BrdU^+^ cells that survived to P60 compared to mice injected on P14 [*t*(30) = 3.99, *p* = 0.001] and P28 [*t*(30) = 5.48, *p* < 0.001]. However, there were no differences observed in the number of remaining BrdU^+^ cells between male WT mice injected on P14 and P28 [*t*(30) = 1.49, *p* = 0.309]. Male ZnT3 KO mice injected on P6 also had a significantly higher number of BrdU^+^ cells that survived to P60 compared to mice injected on P14 [*t*(30) = 4.05, *p* < 0.001] and P28 [*t*(30) = 9.23, *p* < 0.001]. In addition, male ZnT3 KO mice injected on P14 had significantly more BrdU^+^ cells that remained at P60 relative to the mice injected on P28 [*t*(30) = 5.19, *p* < 0.001] (Figure 6A). Together, these results demonstrate a decline in the levels of BrdU^+^ cell survival across the developmental age points at which cells were labelled, with significantly greater survival rates observed for P6-labelled cells than P14- and P28-labelled cells in male WT and ZnT3 KO mice.

For female mice, we found a significant main effect of the genotype [*F*(1, 30) = 13.15, *p* = 0.001] and age [*F*(2, 30) = 82.14, *p* < 0.001]. There was also a significant interaction between the genotype and age [*F*(2, 30) = 7.23, *p* = 0.003], revealing that both female WT and ZnT3 KO mice injected on P6 had significantly greater numbers of BrdU^+^ cells that survived to P60 compared to mice injected on P14 and P28 [WT P6 vs. P14: *t*(30) = 5.02, *p* < 0.001; WT P6 vs. P28: *t*(30) = 5.91, *p* < 0.001; ZnT3 KO P6 vs. P14: *t*(30) = 9.29, *p* < 0.001; ZnT3 KO P6 vs. P28: *t*(30) = 10.88, *p* < 0.001]. There was no difference in the number of BrdU^+^ cells between mice injected on P14 and P28 for both female WT and ZnT3 KO mice [WT P14 vs. P28: *t*(30) = 0.89, *p* = 0.652; ZnT3 KO P14 vs. P28: *t*(30) = 1.59, *p* = 0.265] (Figure 6B). These results demonstrate that, for female WT and ZnT3 KO mice, significantly more BrdU^+^ cells survived to adulthood when the cells were labelled on P6 than P14 and P28.

### 3.5. Cell Proliferation and Survival in Male Mice Injected on P14

Although comparisons between the number of proliferating cells and the number of remaining cells in adulthood cannot technically be compared due to the two being independent processes, an intriguing phenomenon was observed in male ZnT3 KO mice injected on P14. Specifically, we noticed that the number of BrdU^+^ cells between the male ZnT3 KO proliferation and survival groups were very similar (Appendix A). As such, we conducted exploratory comparisons and found that, while the number of BrdU^+^ cells labelled at P14 was significantly higher than the number of BrdU^+^ cells that were remaining at P60 in male WT mice [*t*(5.13) = 3.80, *p* = 0.012], no significant differences were found in male ZnT3 KO mice [*t*(10) = 0.42, *p* = 0.682] (Figure 7A). We also calculated the ratio of the number of BrdU^+^ cells at P60 to the number of BrdU^+^ cells 24 h after the last BrdU injection on P14 and found that 18.37% of cells labelled at P14 survived to adulthood in WT mice, whereas ZnT3 KO mice had a 90.61% cell survival ratio (Figure 7B). These results suggest that there was a sustained survival of P14-labelled cells in male ZnT3 KO mice.

### 3.6. Neuronal Phenotype of Surviving Cells in Mice Injected on P6, P14, and P28

We next examined the neuronal phenotype of cells that survived to P60 in the SGZ/GCL of male and female WT and ZnT3 KO mice that received BrdU injections on P6, P14, and P28 (Figure 8A′″). The results revealed there was a significant main effect of age [*F*(1, 24) = 23.86, *p* < 0.001], with higher percentages of BrdU^+^ cells co-labelled with NeuN observed in mice injected on P6. This effect did not differ based on the genotype [*F*(2, 24) = 2.29, *p* = 0.123] or sex [*F*(2, 24) = 0.36, *p* = 0.703]. There was also a significant main effect of sex [*F*(1, 24) = 13.25, *p* = 0.001], with significantly higher percentages of BrdU^+^ cells co-labelled with NeuN observed in female mice overall compared to male mice. While this effect did not differ based on genotype [*F*(1, 24) = 0.001, *p* = 0.972], this effect was only statistically significant in mice injected on P14, with 82.77% of BrdU^+^ cells also immunoreactive for NeuN in female mice, compared to 75.37% of BrdU^+^ cells immunoreactive for NeuN in male mice [*F*(1,24) = 7.79, *p* = 0.010] (Figure 8B). There was no main effect of the genotype [*F*(1, 24) = 0.19, *p* = 0.665], nor was there a three-way interaction between the sex, genotype, and age [*F*(2, 24) = 0.48, *p* = 0.625] (Appendix A).

## 4. Discussion

### 4.1. Cell Proliferation in Mice Injected on P6, P14, and P28

#### 4.1.1. Cell Proliferation in Mice Injected on P6

When we assessed cell proliferation at P6 in WT and ZnT3 KO mice, we found no differences between the two genotypes, in either male or female mice. These results suggest that vesicular zinc, or a lack thereof, may not be critically involved in impacting the genesis of cells at the P6 age point. Our results conflict with previous studies describing reduced proliferation of neural stem cells in the embryonic and postnatal rat brains resulting from gestational maternal zinc deficiency [46,47]. However, this discrepancy could be due to these previous studies utilising maternal zinc deficiency, which systemically reduces zinc levels. Thus, compared to the ZnT3 KO mice that only have vesicular zinc in the brain eliminated, the bioavailability of zinc throughout the entire body is restricted, which could affect a greater number and variety of proliferation-related pathways [73]. Accordingly, these effects in multiple different pathways can impair normative cellular functioning, resulting in decreased cell proliferation and survival of cells [7,37]. Thus, the systemic loss of zinc may have greater consequences on cell proliferation at P6 than the loss of vesicular zinc, given the more global effects of the former.

#### 4.1.2. Cell Proliferation in Mice Injected on P14

When we examined cell proliferation at P14 in WT and ZnT3 KO mice, we found that male ZnT3 KO mice had significantly lower numbers of BrdU^+^ cells compared to WT mice. There were no differences in cell proliferation at P14 between female WT and ZnT3 KO mice. These results suggest that a loss of vesicular zinc influences cell proliferation at P14 and that this effect is specific to males. A potential explanation for why we observed decreased numbers of BrdU^+^ cells in male ZnT3 KO mice is the pathways that vesicular zinc can modulate. Of particular interest is the extracellular signal-regulated kinases (ERK1/2) signalling pathway, which regulates a wide variety of cellular processes central to brain development, including cell proliferation, differentiation, and survival [74]. Previous research has demonstrated that the genetic deletion of ZnT3 in mice results in suppressed cell proliferation due to reduced levels of ERK phosphorylation (pERK) [36]. Further research should aim to elucidate the mechanisms that underlie the dysregulation of ERK1/2 associated with the elimination of vesicular zinc and how it contributes to altered brain development, structure, and function.

Interestingly, this reduction of cell proliferation at P14 was not seen in female ZnT3 KO mice, suggesting that cell proliferation is uniquely disrupted in male ZnT3 KO mice at this specific age. These alterations in cell proliferation can have downstream consequences for cell differentiation, neuron maturation, and synaptic function, all of which are implicated in certain neurodevelopmental disorders. One such disorder includes autism spectrum disorder (ASD), which has been suggested to result from perturbations during brain development that disrupt cell proliferation; differentiation; and the formation, maturation, and maintenance of synapses [75]. Previous studies have shown that male ZnT3 KO mice display behavioural features that are characteristic of ASD, such as reduced social interaction and increased repetitive behaviour, which were accompanied by abnormalities in the neurogenesis and brain size [17]. However, contrary with our results, these abnormalities consisted of a greater number of neurons in the frontal cortex in tandem with an enlargement in brain size, particularly in the frontal region [17]. It is possible that our findings differ from these previous results because we only examined cell proliferation in the hippocampus and did not account for differences in brain size. Regardless, this link between vesicular zinc, altered cell proliferation, and ASD may explain why we only observed a significant effect in males, as ASD has a higher male preponderance bias [76]. Examining whether this discrepancy exists, how it is linked to the quantity of neurons in different regions of the brain, and the functional connectivity of these neurons will be essential in elucidating the link between vesicular zinc and ASD.

#### 4.1.3. Cell Proliferation in Mice Injected on P28

At P28, we did not observe any differences in cell proliferation between WT and ZnT3 KO mice in either the male or female mice, suggesting that vesicular zinc, or a lack thereof, may not be critically involved in influencing cell proliferation at the P28 age point. Nevertheless, our results are consistent with previous unpublished experiments from our laboratory, where we showed that a loss of vesicular zinc does not affect the levels of the baseline neural proliferation in adult mice under standard conditions [43,77]. Only under non-standard conditions where mice are exposed to positive modulators such as environmental enrichment or chronic fluoxetine treatment is the loss of vesicular zinc made evident [43,77]. The standard housing conditions in which our P28 mice were living in may explain the lack of differences in cell proliferation between genotypes. Overall, these results emphasise the notion that vesicular zinc may not be necessary for maintaining cell proliferation, at least under normal conditions, in mice that are P28 and older. Further investigation into different modulators of hippocampal neurogenesis—both positive and negative—at different stages of development will yield valuable information about vesicular zinc’s role in mediating experience-dependent plasticity across the lifespan.

Though we were unable to detect an effect of eliminating ZnT3 on the cell proliferation levels, we did find a sex difference, whereby male mice had approximately 43% more proliferating cells than female mice. Our results are consistent with those of Siddiqui and Romeo [78], who found that pre-adolescent (P30) male rats exhibited approximately 40% more proliferating cells in the hippocampus than pre-adolescent females. It is worth highlighting that the results of the current study may reflect novel findings, as most studies documenting sex differences in hippocampal neurogenesis have focused primarily on the adulthood period. In comparison, relatively less is known about sex differences in the developmental stages before that. Furthermore, there are variations between the findings regarding sex differences in hippocampal neurogenesis due to species and strain differences [79,80,81], rendering it difficult to directly compare our results with those in the literature. As such, the interpretation of our results remains largely speculative. Examining cell proliferation between males and females across multiple stages of the lifespan, including during the early postnatal period and pre-, mid-, and late adolescence, as well as in adulthood, would provide stronger evidence for transient sex differences in the developmental progression of hippocampal neurogenesis.

Another factor that may underlie the sex difference in males and females is the interaction between ZnT3 expression and oestrogen. It has been demonstrated that oestrogen replacement reduces ZnT3 expression and, thus, vesicular zinc levels in the brain, whereas ovariectomy increases the levels of ZnT3 and vesicular zinc [82]. Therefore, it is possible that this downregulation of ZnT3 expression and levels of vesicular zinc in the female WT mice results in decreased cell proliferation [82]. If this were the case, it would also indicate that the level of cell proliferation observed in female WT mice is proportionate to the level of cell proliferation in ZnT3 KO mice, which would explain why we did not observe any differences between the two genotypes. An in-depth analysis of the oestrogen and ZnT3 levels in the brain will be required to conclusively determine whether the sex difference observed in the proliferation of cells at P28 are due to an effect of oestrogen.

### 4.2. Cell Survival in Mice Injected on P6, P14, and P28

#### 4.2.1. Cell Survival in Mice Injected on P6

When we examined the number of cells labelled on P6 that were remaining in adulthood, we found that both male and female ZnT3 KO mice retained significantly more cells compared to WT mice. It is interesting that we observed greater levels of cell survival in ZnT3 KO mice injected on P6 than WT mice, as most of the literature regarding zinc and cell survival indicate that it is the loss of zinc that impairs cell survival through the modulation of several pro-survival and proapoptotic pathways that can induce apoptosis via the intrinsic pathway [83,84]. For instance, inhibition of the pro-survival pathways ERK and Nuclear factor-kappa B (NF-κB), as well as activation of Caspase-3, a cysteine protease that is central in programmed cell death [85], has been found to be associated with low zinc levels [83,84]. Thus, our findings that ZnT3 KO mice exhibited greater cell survival conflict with previous studies. However, the previous research mentioned utilised dietary zinc deficiency models in adult rats [83] and in vitro analyses of human neuroblastoma IMR-32 cells and primary cultures of rat cortical neurons [84] and, as a result, may reflect the biological role of zinc rather than the role of zinc in synaptic signalling. Our results may also indicate that zinc-mediated mechanisms of cell survival and apoptosis for cells born during the early postnatal developmental period may differ from those born in adulthood. For instance, the balance between activation and inhibition of pro-survival and proapoptotic pathways may favour greater activation of pro-survival pathways in ZnT3 KO mice for cells labelled on P6, resulting in greater cell survival, whereas this balance may shift towards greater activation of proapoptotic pathways for cells born at later stages in the lifespan, including adulthood. Examining the different mechanisms and molecular events involved in the maintenance and elimination of newly generated cells during development will be required to provide more conclusive insight into the process of cell retention in the absence of vesicular zinc.

#### 4.2.2. Cell Survival in Mice Injected on P14

When we assessed the survival of cells at P60 in mice injected with BrdU on P14, we found that male ZnT3 KO mice retained significantly more cells compared to male WT mice. This finding was specific to male mice, as female ZnT3 KO and WT mice injected with BrdU on P14 did not display any differences in the number of cells that survived to P60. Additionally, we also found that male mice injected on P14 had significantly higher levels of cell survival compared to female mice, supporting our hypothesis that there would be differences between male and female mice. As previously discussed, the increased cell survival observed in male ZnT3 KO mice may be the resultant of the balance between the activation and inhibition of pro-survival and proapoptotic pathways. Additionally, why we do not see this same effect in female mice could be attributed to the downregulation of ZnT3 levels by oestrogen and, thus, decreased levels of vesicular zinc in female WT mice, rendering a lack of difference in cell retention between genotypes [82].

#### 4.2.3. Cell Survival in Mice Injected on P28

In male and female mice injected on P28, there were no differences in the number of cells that survived to P60 between WT and ZnT3 KO mice, suggesting that a lack of vesicular zinc may not be critically involved in affecting the survival of cells born at the P28 developmental age point. These results are consistent with the unpublished experiments from our laboratory previously discussed where Chrusch [43] and Boon [77] showed that the enhancement of hippocampal cell proliferation and survival under conditions of enrichment is vesicular zinc-dependent, whereas vesicular zinc may not be necessary for maintaining cell proliferation and survival under normal, standard conditions. Given our mice were administered BrdU on P28 and left in their standard home cages until P60 to assess cell survival, it is possible that the lack of positive modulators, such as being housed in an enriched environment, explains the lack of differences seen in cell survival between WT and ZnT3 KO mice. Further investigation into whether positive modulators affect the survival of cells labelled at P28 will be necessary to substantiate this postulation.

Interestingly, it should be noted that there were no sex differences observed in the survival of BrdU^+^ cells that were labelled at P28 when assessed at P60, despite our previous findings that male mice had greater levels of cell proliferation compared to female mice at this age (see Section 4.1.3). Although comparisons between the number of cells that were labelled at P28 and the number of cells that were remaining at P60 were not conducted, we can speculate that, although both male and female mice showed a reduction in the number of cells between labelling at P28 and assessment of the remaining cells at P60, this decrease may be more pronounced in male mice, as males exhibited higher levels of BrdU^+^ cells at P28 compared to females. This could further be interpreted as male mice showing a greater attrition of cells born at P28 or, alternatively, male mice retaining less BrdU^+^ cells. This interpretation would fit with the findings from a previous study demonstrating that males produce more new cells and have greater cell attrition, whereas females produce relatively less new cells, which are maintained throughout maturation, though a caveat of this study is that adult (P60) rats were used [57]. In any case, this raises the intriguing question of why males display significantly higher levels of cell proliferation compared to females yet the numbers of cells that survived into adulthood were not significantly different. One interpretation may be that the greater number of new cells generated at P28 in males is a consequence of greater cell overproduction relative to what is seen in females [86]. This may result in a greater subset of dispensable cells in males that needs to be eliminated, potentially explaining the previous finding of why males show a greater attrition of newly generated cells [57,86]. While females also undergo this active elimination process, perhaps they have fewer dispensable cells. This might also suggest that a majority of cells generated and labelled at P28 in females are the same cells that survive to P60, indicating these cells are presumably the important, indispensable ones. If this were the case, it is also possible that the cells that were retained in males are the same type of cells that were retained in females, explaining the lack of sex differences in cell survival. It would be important to further examine these sex differences in the regulation of cell proliferation, maturation, and the trajectory of cell survival across different timepoints of the lifespan, as well as how vesicular zinc modulates these processes. Furthermore, future studies should aim to characterise these cells, explore the mechanisms underlying these differences, and determine whether there is a functional implication of these differences.

### 4.3. Cell Proliferation and Survival in Male Mice Injected on P14

A remarkable observation in male ZnT3 KO mice injected with BrdU on P14 was the direction of the differences in cell proliferation and cell survival. When we examined cell proliferation, we found that male ZnT3 KO mice had significantly fewer proliferating cells than male WT mice. It is interesting, then, that when we examined cell survival in adulthood, male ZnT3 KO mice had significantly more cells that survived compared to male WT mice. Moreover, there were no differences in the number of BrdU^+^ cells labelled at P14 and the number of BrdU^+^ cells that remained at P60. In fact, 90.61% of the cells that were labelled at P14 survived to adulthood. In contrast, male WT mice had a 18.37% survival rate. One interpretation for this observation is that, given the low levels of cell proliferation to start with, there was a lower proportion of cells that could have been eliminated due to them being indispensable, and as a result, most of them survived. Alternatively, the lack of a difference between the number of proliferating cells at P14 and the number of remaining cells at P60 may reflect aberrant cell pruning in male ZnT3 KO mice. The elimination of cells through apoptosis and autophagic cell death are key processes that are essential for the development of normal brain growth and balanced networks, and alterations in these processes can subsequently affect synaptic pruning, a process in which weak connections are targeted and eliminated in favour of stronger ones [83,87]. The dysregulation of synaptic pruning can ultimately result in excess synapses and synaptic connections, which have been implicated in neurodevelopmental disorders, including ASD [87,88]. This interpretation fits with our previous finding and explanation whereby decreased cell proliferation, in addition to atypical cell survival, point towards abnormal processes during brain development due to a lack of vesicular zinc, which might underlie the pathogenesis of ASD. This would also help explain why we observe this effect in male but not female mice given the high male prevalence of ASD. Importantly, caution is warranted when interpreting these observations, because different processes involved in cell proliferation and cell survival do not allow us to directly compare the two. Regardless, further investigation into how vesicular zinc affects different neurobiological mechanisms involved in cell proliferation and survival and the morphology of different cell types, as well as other regions of the brain known to be affected in ASD, will help to clarify our findings and the link between vesicular zinc and ASD.

### 4.4. Age Effects on Cell Proliferation in Mice Injected on P6, P14, and P28

We additionally assessed whether there were any effects of the age at which BrdU was injected on cell proliferation throughout the early developmental period. In male WT mice, we observed that, surprisingly, the levels of cell proliferation did not differ across the three ages at which BrdU was administered. In contrast, male ZnT3 KO mice showed the lowest levels of cell proliferation at P14. These results were surprising, as it has long been established that neurogenesis rates change across the lifespan. Moreover, our results conflict with the standard progression of neurogenesis, in which there is a peak observed around birth, followed by an exponential decline [72]. It is possible that we failed to detect changes due to selecting postnatal ages that do not capture the peak and decline of neurogenesis. For instance, it may be the case that the peak of neurogenesis occurs closer to P0, and the decline occurs very rapidly after. As a result, we may be observing more stable levels of moderate cell proliferation across P6, P14, and P28. This may be especially true given we also see similar levels of cell proliferation in ZnT3 KO mice across P6 and P28, and cell proliferation at these ages does not differ when compared to WT mice.

Regarding the dramatic reduction in cell proliferation seen in ZnT3 KO mice injected on P14, this may reflect age-specific consequences of eliminating vesicular zinc, as previously discussed (see Section 4.1.2). An additional, and not mutually exclusive, suggestion is that P14 represents a particularly unique developmental age point for male ZnT3 KO mice wherein the loss of vesicular zinc may have more notable effects on cell proliferation. This raises many interesting questions, including whether this implies that cell proliferation in male ZnT3 KO mice is more susceptible to modulators of experience-dependent plasticity. Perhaps male ZnT3 KO mice experience standard housing conditions as impoverished conditions, leading to decreases in neurogenesis due to this environmental deprivation. If this were the case, it is possible that being housed in an enriched environment would increase the cell proliferation in ZnT3 KO mice up to the levels of WT mice. Furthermore, since cell proliferation at P14 did not differ between female WT and ZnT3 KO mice, this begs the question of whether this unique developmental age point is absent in female mice. Alternatively, it may be that male and female mice have different developmental trajectories whereby one sex matures faster than the other, in which case, the P14 age point in male mice may align with a completely different postnatal day in female mice. Moreover, this prompts further questions about whether this effect is only present on P14 in male mice or whether it spans across multiple postnatal days. Future research delineating the postnatal day(s) of when this effect is present in male and female mice will be required to address these questions.

When we examined female mice, we found that both female WT and ZnT3 KO mice had higher levels of BrdU^+^ cells at P6 and P14 compared to P28, though the difference between P14 and P28 in ZnT3 KO mice was not significant. Unlike male mice, these results illustrate how neurogenesis during development changes in an age-dependent manner, which is consistent with the general pattern of neurogenesis observed across the lifespan in mice: early in life around birth, a robust peak in neurogenesis is observed, followed by a decline that prefaces the beginning of a protracted period of low-level neurogenesis, which spans the remainder of the lifespan [72,89]. It would be worthwhile to examine the cellular properties underlying the proliferation of cells at different age points, especially considering we saw different patterns in male and female mice.

### 4.5. Age Effects on Cell Survival in Mice Injected on P6, P14, and P28

When we assessed whether there were any effects of the age at which BrdU was injected on the survival of cells as assessed at P60, we found that there were higher levels of cells that survived in both male WT and ZnT3 KO mice that were injected on P6 than P14 and P28. Male mice injected on P14 also had higher levels of the remaining cells compared to mice injected on P28, though this was not significant in WT mice. These results suggest that a greater number of cells that were born and labelled during the first postnatal week survived. The differences in the survival of cells born at different age points may reflect the differences in their functional role in the hippocampus. For instance, the demands during the periods of rapid brain development may require that higher levels of cells born on P6 are retained during the initial few weeks and months, while relatively lower levels of survival for cells born during the later developmental period may be sufficient for hippocampal functions [90]. There is evidence to suggest that the elimination of neurons during the juvenile, adolescent, and early adulthood periods is necessary to promote plasticity, which is crucial for the acquisition of certain types of contextual memory, such as pattern separation [91,92]. Thus, lower survival levels for cells born on P14 and P28 may in fact be beneficial, which could explain why we see lower levels of BrdU^+^ cells surviving to P60 in mice injected on those days. Further examination of how the levels of neurogenesis differ throughout development and the functional implications of such changes will be necessary to clarify our findings.

In terms of female mice, both female WT and ZnT3 KO mice injected on P6 had greater numbers of BrdU^+^ cells that survived to P60 compared to mice injected on P14 and P28. As described above, differences in the survival of age-specific cohorts of cells may suggest that cells born at different ages during the early postnatal period are distinct in that they have unique functional roles. Further investigation into the interplay between the addition, survival, and removal of neurons is therefore warranted, particularly if we want to understand the functional relevance of new neurons.

### 4.6. Neuronal Phenotype of Surviving Cells in Mice Injected on P6, P14, and P28

We also examined the neuronal phenotypes of cells that survived to P60 in mice injected with BrdU on P6, P14, and P28 and found no differences in the percentages of BrdU^+^ cells co-expressing NeuN between WT and ZnT3 KO mice in either male or female mice at any of the age points, indicating that neuronal differentiation is not affected by the lack of vesicular zinc. This was interesting, as we previously showed that male and female ZnT3 KO mice injected on P6, as well as male ZnT3 KO mice injected on P14, had significantly more BrdU^+^ cells that survived compared to the WT mice. This indicates that differences in the cell survival numbers do not necessarily foretell differences in the cell phenotypes. It is worth noting that the proportion of P14-labelled cells that differentiated into mature neurons in male ZnT3 KO mice (78.67%) was higher than WT mice (72.08%), although this was not statistically significant. Overall, these findings are consistent with previous findings from our laboratory, which showed no differences in neuronal differentiation between WT and ZnT3 KO mice under standard housing conditions [43].

We also found that the percentage of BrdU^+^ cells co-expressing NeuN was significantly higher for P6-labelled cells in male and female WT and ZnT3 KO mice compared to P14- and P28-labelled cells, which may reflect age-dependent differences in the phenotypic differentiation of new-born cells into neurons throughout the lifespan. Analyses of neuronal phenotype percentages for P14- and P28-labelled cells were in the range previously reported for adult mice [93,94] and, thus, suggests that P6 may be more representative of a “developmental” age. In fact, some studies suggest that “adult” neurogenesis may begin as early as P14, when the dentate gyrus more closely resembles neurogenesis and the anatomical layout of the adult brain [69]. In future studies, it will be important to further examine these differences in the differentiation of newly generated cells, especially in the context of the functionality of these new neurons.

Interestingly, we observed a sex difference, such that a greater percentage of BrdU^+^ cells were co-localised with NeuN in female mice compared to male mice, although this was only statistically significant for mice injected on P14. This contrasts our previous finding in which significantly more BrdU^+^ cells survived to adulthood in male mice injected on P14 compared to female mice. These findings suggest that, within female mice, despite fewer P14-labelled BrdU^+^ cells surviving, there is a higher percentage of these cells that differentiate into neurons. To our knowledge, neither of these findings have been reported previously, and the mechanisms underlying the sex differences in neuronal differentiation remain to be determined. Thus, it will be important for future research to focus on the possible mechanisms underlying these sex differences, as well as other factors involved in influencing different neurogenic stages, including cell proliferation, survival, differentiation, maturation, and integration.

## 5. Summary and Conclusions

The primary goal of the present experiment was to examine whether vesicular zinc is involved in modulating cell proliferation, cell survival, and neuronal differentiation in the developing hippocampus. To do this, we evaluated cell proliferation at three different developmental age points (P6, P14, and P28) in male and female ZnT3 KO mice that lack vesicular zinc. A secondary goal of the current study was to examine whether vesicular zinc is involved in modulating the survival or neuronal differentiation of cells. To accomplish this, we evaluated the survival of cells that were labelled at P6, P14, and P28 in adulthood (P60), as well as the neuronal phenotype of these surviving cells. Finally, a tertiary goal of the current study was to explore whether there were any differences in cell proliferation, cell survival, and/or neuronal differentiation based on sex and developmental age points. We showed that male ZnT3 KO mice have reduced cell proliferation at P14, but the survival of these cells are sustained to P60. Both male and female ZnT3 KO mice display greater levels of survival for cells labelled at P6. In terms of sex differences, we found that, at P28, male mice had higher levels of cell proliferation compared to female mice. Male mice also showed greater levels of cell survival for P14-labelled cells. However, female mice showed higher percentages of BrdU/NeuN co-localisation in P14-labelled cells. Regarding age effects, the lowest level of cell proliferation was at P14 for male ZnT3 KO mice, while female mice, regardless of genotype, displayed the lowest level of cell proliferation at P28. For cell survival, the numbers of cells that were retained steadily decreased based on the age of the BrdU injection, with the greatest levels of cell survival and neuronal differentiation seen for cells labelled at P6 for all mice, regardless of genotype or sex. These results are suggestive of a potential role for vesicular zinc in modulating hippocampal cell proliferation, survival, and neuronal differentiation during the developmental period and demonstrate sex- and age-dependent differences. While these findings are intriguing, additional research will be required to examine the mechanisms through which vesicular zinc could be exerting its effects on these neurogenic stages during development. Overall, our results should be regarded as a starting point that provides the foundation for future studies to examine the role of vesicular zinc in developmental hippocampal neurogenesis.

## Figures and Tables

**Figure 1 cells-12-00880-f001:**
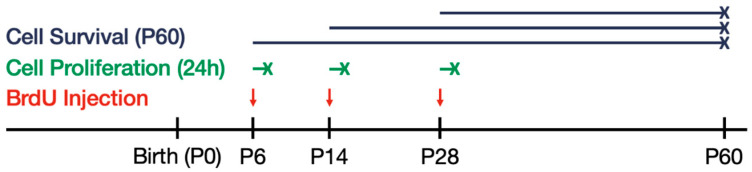
Experimental timeline of bromodeoxyuridine (BrdU) injections and endpoints used to assess cell proliferation and survival. Male and female WT and ZnT3 KO mice (*n* = 144) were assigned to postnatal day (P) 6, 14, or 28; at which point, two subcutaneous injections of BrdU (50 mg/kg) were administered at approximately 09:00 and 11:00 h (indicated by red arrow). To assess the cell proliferation, half of the mice from each group were killed approximately 24 h after the last injection (indicated by green -x). The other half were left to survive until P60 to assess the cell survival (indicated by blue -x).

**Figure 2 cells-12-00880-f002:**
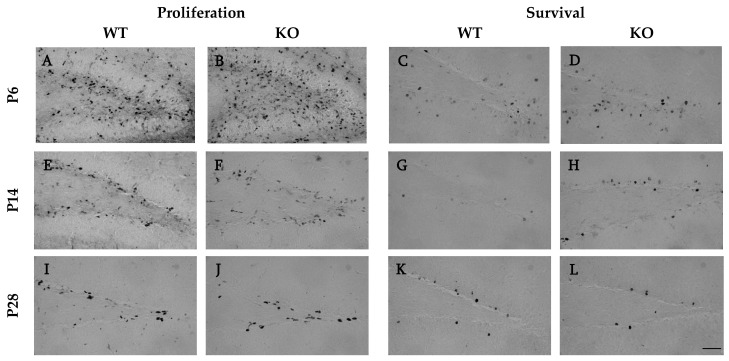
Photomicrographs of diaminobenzidine (DAB)-stained BrdU^+^ cells in the dentate gyrus of the hippocampus of animals injected with BrdU. Representative sagittal sections from a (**A**) WT and (**B**) ZnT3 KO mouse in the P6 proliferation group, (**C**) WT and (**D**) ZnT3 KO mouse in the P6 survival group, (**E**) WT and (**F**) ZnT3 KO mouse in the P14 proliferation group, (**G**) WT and (**H**) ZnT3 KO mouse in the P14 survival group, (**I**) WT and (**J**) ZnT3 KO mouse in the P28 proliferation group, and (**K**) WT and (**L**) ZnT3 KO mouse in the P28 survival group. All photomicrographs were taken at 40× magnification. Scale bar = 50 µm.

**Figure 3 cells-12-00880-f003:**
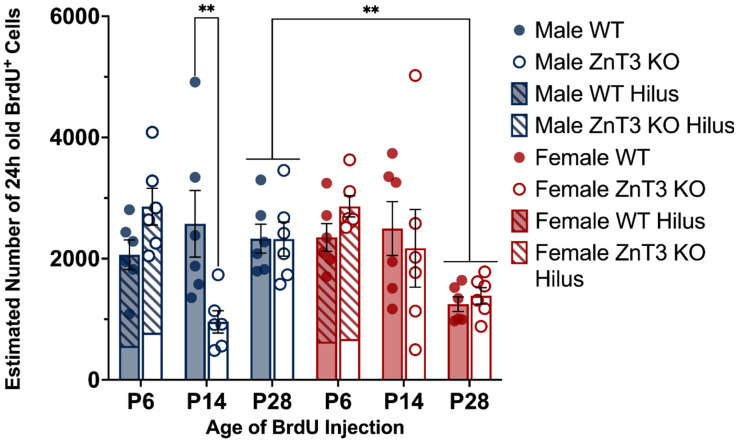
Proliferation of cells labelled at P6, P14, and P28, as measured by BrdU^+^ cells, in male and female WT and ZnT3 KO mice. BrdU was administered to male and female WT and ZnT3 KO mice on P6, P14, or P28 to label dividing cells (*n* = 6 per group). To assess cell proliferation, brains were collected approximately 24 h after the last injection, and BrdU^+^ cells were counted in the hilus and SGZ/GCL for P6 brains (P6 bars represent the aggregate; diagonal stripes indicate counts from the hilus) and the SGZ and GCL for P14 and P28 brains. Male WT mice injected on P14 had significantly higher numbers of BrdU^+^ cells compared to male ZnT3 KO mice. Male mice injected on P28, regardless of genotype, had significantly higher numbers of BrdU^+^ cells compared to female mice. Error bars represent ± standard error of the mean. ** Indicates *p* < 0.010.

**Figure 4 cells-12-00880-f004:**
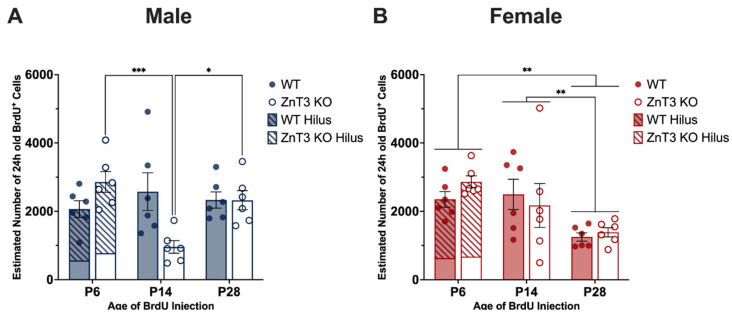
Age effects on cell proliferation, as measured by BrdU^+^ cells, in male and female WT and ZnT3 KO mice injected on P6, P14, and P28. BrdU was administered to male and female WT and ZnT3 KO mice on P6, P14, or P28 to label dividing cells (*n* = 6 per group). To assess cell proliferation, brains were collected approximately 24 h after the last injection, and BrdU^+^ cells were counted in the hilus and SGZ/GCL for P6 brains (P6 bars represent the aggregate; diagonal stripes indicate counts from the hilus) and the SGZ and GCL for P14 and P28 brains. (**A**) There were no significant differences in the number of BrdU^+^ cells between male WT mice injected on P6, P14, and P28. In contrast, male ZnT3 KO mice injected on P14 had significantly lower levels of BrdU^+^ cells compared to mice injected on P6 and P28. (**B**) In female mice, WT and ZnT3 KO mice injected on P6 and P14 had significantly higher levels of BrdU^+^ cells compared to mice injected on P28. Error bars represent ± standard error of the mean. * Indicates *p* < 0.050; ** indicates *p* < 0.010; *** indicates *p* < 0.001.

**Figure 5 cells-12-00880-f005:**
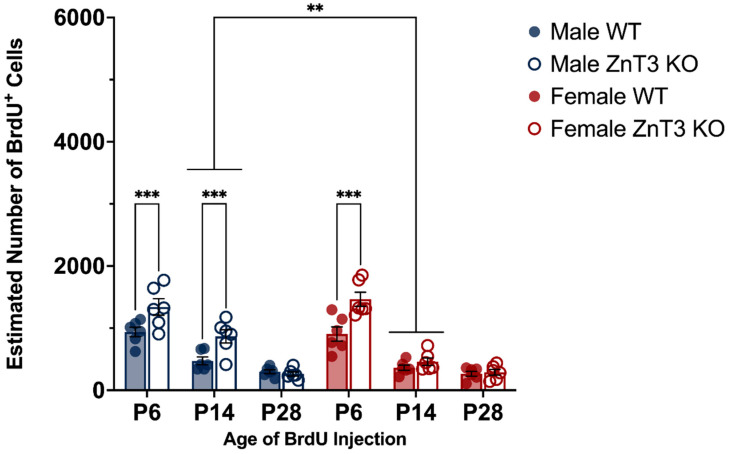
Survival of cells labelled at P6, P14, and P28, as measured by BrdU^+^ cells at P60, in male and female WT and ZnT3 KO mice. BrdU was administered to male and female WT and ZnT3 KO mice on P6, P14, or P28 to label dividing cells (*n* = 6 per group). To assess the cell survival, brains were collected on P60, and BrdU^+^ cells were counted in the SGZ and GCL. Male ZnT3 KO mice injected on P6 and P14 had significantly higher numbers of BrdU^+^ cells that survived to P60 compared to male WT mice. Female ZnT3 KO mice injected on P6 also had significantly more BrdU^+^ cells remaining at P60 compared to female WT mice. Male mice injected on P14, regardless of genotype, had significantly higher numbers of BrdU^+^ cells that remained at P60 compared to female mice. Error bars represent ± standard error of the mean. ** Indicates *p* < 0.010; *** indicates *p* < 0.001.

**Figure 6 cells-12-00880-f006:**
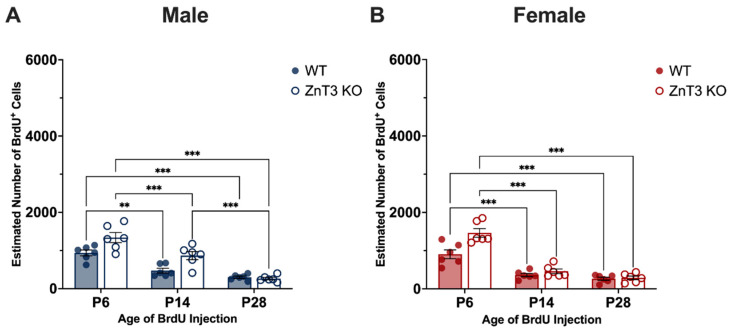
Age effects on cell survival, as measured by BrdU^+^ cells at P60, in male and female WT and ZnT3 KO mice injected on P6, P14, and P28. BrdU was administered to male and female WT and ZnT3 KO mice on P6, P14, or P28 to label dividing cells (*n* = 6 per group). To assess the cell survival, brains were collected on P60, and BrdU^+^ cells were counted in the SGZ and GCL. (**A**) Male WT mice injected on P6 had significantly higher numbers of BrdU^+^ cells that remained at P60 compared to male WT mice injected on P14 and P28. Similarly, male ZnT3 KO mice injected on P6 also had significantly higher numbers of BrdU^+^ cells that remained at P60 compared to male ZnT3 KO mice injected on P14 and P28. In addition, higher levels of BrdU^+^ cell survival were also observed in male ZnT3 KO mice injected on P14 relative to mice injected on P28. (**B**) Female WT and ZnT3 KO mice injected on P6 displayed significantly higher levels of BrdU^+^ cell survival at P60 compared to mice injected on P14 and P28. Error bars represent ± standard error of the mean. ** Indicates *p* < 0.010; *** indicates *p* < 0.001.

**Figure 7 cells-12-00880-f007:**
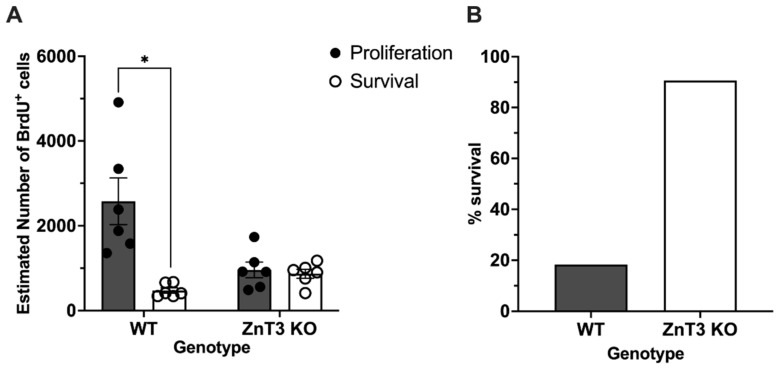
Proliferation and survival of cells labelled at P14, as measured by BrdU^+^ cells, in male WT and ZnT3 KO mice. BrdU was administered to male WT and ZnT3 KO mice on P14 to label dividing cells (*n* = 6 per group). To assess cell proliferation, brains were collected approximately 24 h after the last injection. To assess cell survival, brains were collected on P60. BrdU^+^ cells were counted in the SGZ and GCL. (**A**) Male WT mice had significantly higher numbers of BrdU^+^ cells at P14 compared to at P60. In contrast, the number of BrdU^+^ cells at P14 and P60 did not differ for male ZnT3 KO mice. (**B**) In male WT mice, 18.37% of cells labelled at P14 survived to P60. In male ZnT3 KO mice, 90.61% of cells labelled at P14 survived to P60. Error bars represent ± standard error of the mean. * Indicates *p* < 0.050.

**Figure 8 cells-12-00880-f008:**
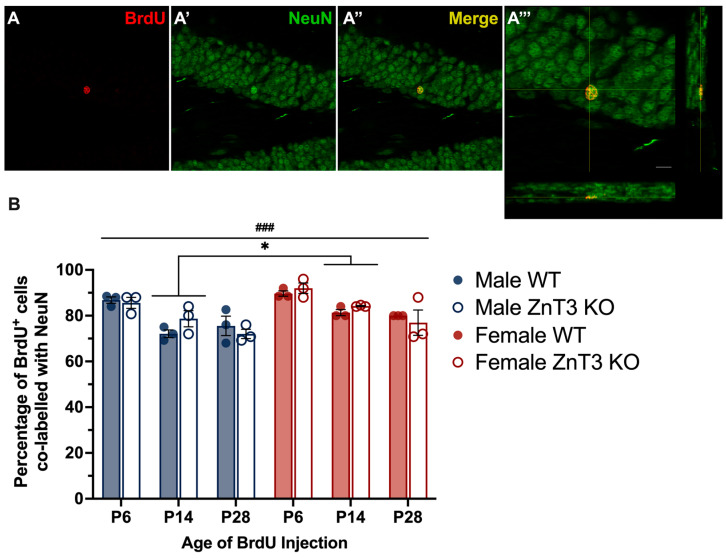
Neuronal phenotype of cells that survived to P60 in male and female WT and ZnT3 KO mice injected on P6, P14, and P28. BrdU was administered to male WT and ZnT3 KO mice on P6, P14, or P28 to label dividing cells. Brains were collected on P60, and a subset of brain sections was stained for BrdU and NeuN (*n* = 3 per group). Representative confocal microscopic images of (**A**) BrdU- (red) and (**A′**) NeuN-immunopositive cells (green) in the same plane. (**A″**) Neuronal phenotype of the cells was determined by co-localisation of BrdU and NeuN (yellow), as shown in the (**A′″**) representative image with an orthogonal view (BrdU in red co-labelled with NeuN in green shows as yellow). All photomicrographs were taken at 60× magnification. Scale bar = 10 µm. (**B**) While female mice had a significantly higher percentage of BrdU^+^ cells that were co-localised with NeuN compared to male mice across all ages, this was only statistically significant for cells labelled on P14. Cells that were labelled on P6 also had significantly higher percentages of BrdU^+^ cells that were co-localised with NeuN compared to cells labelled on P14 and P28. Error bars represent ± standard error of the mean. ^###^ Main effect of age, *p* < 0.001; * indicates *p* < 0.050.

## Data Availability

The data presented in this study are available on request from the corresponding author upon reasonable request.

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
