# Peer review of "Vesicular Zinc Modulates Cell Proliferation and Survival in the Developing Hippocampus"

_cells, 2023, doi:10.3390/cells12060880_

Round 1

Reviewer 1 Report (Previous Reviewer 2)

The authors have adequately responded to my critiques. 

Reviewer 2 Report (Previous Reviewer 1)

Authors addressed main question related to the identity of the cell type that they measured in the WT and ZnT3 KO mice. I do not have any further questions on the manuscript.

This manuscript is a resubmission of an earlier submission. The following is a list of the peer review reports and author responses from that submission.

Round 1

Reviewer 1 Report

Fu et al. studied the role of vesicular zinc in modulating neurogenesis during the early postnatal period using WT and ZnT3 KO mice. And found reduced cell proliferation in ZnT3 KO mice at P14, which exhibited more remarkable cell survival in contrast to WT at P60. Authors also highlighted the sex-dependent effects on hippocampal neurogenesis in the developing brain. The study cohort is well-designed, with an adequate number of mice per group. The paper is well written with a precise aim in mind. However, my primary concerns are that the paper is very descriptive and lacks some mechanistic insights, especially in light of their findings. Does ZnT3 mRNA or protein differentially expressed in males vs. females in the hippocampus? Also, it is a bit strange to see that all the labeled cells in the ZnT3 KO male mice (~1000 cells) survived (~1000 cells). Whereas for all other mice, there is a 50% difference between labeling and survival. Lastly, their main goal is to check whether vesicular zinc modulates hippocampal neurogenesis. BrdU doesn’t discriminate between neurons and glia. Hence whether the labeled cells are neurons or glia is still not answered. Apart from these significant comments, I also have minor remarks to improve the manuscript.

Minor comments:

-       In the abstract rationale for labeling cells at P6 or P14 is lacking. Hence at first glance, it is very confusing to read as many time points were mentioned. Authors should consider simplifying the abstract while keeping enough details.

-       Authors mention in the introduction line 64, “neurogenesis, cell survival, and learning and memory abilities including that of spatial memory in rodents are strongly enhanced when housed in an enriched environment” it would be nice to elaborate on specific enrichment methods they are referring to. Is it a running wheel?

-       In figure 3: for the P6 timepoint Hilus region is considered while SGZ and GCL regions were measured for P14 and P28 brains. An explanation must be provided for this discrepancy.

-       Data from the immunohistochemistry and figures 3 and 4 are not matching. At p14, the P60 WT panel looks like fewer cells than the ZnT3KO, while Figures 3 and 4 have the opposite effect.

-       Figure 3 and Figure 4 data look very identical. If it is from the same experiment, is it necessary to represent it twice?

-       The same is the case for Figures 5 and 6.

-       At P14, BrdU-positive cells are lower in the ZnT3 KO mice. Given the P6 and P28 labeling is usual, how do authors explain this? Did they check whether there is any technical problem with injections?

-       Are these BrdU-labeled cells neurons or glia?

Reviewer 2 Report

The study by Fu etal offers interesting information on the role of zinc transporter 3 in postnatal neurogenesis. The study is well conducted and well written. There are a few issues the authors need to address.

Methods: Brains were cut along the midsagittal axis and one hemisphere was used for sections. It is not clear if the same hemisphere was used in all the animals, and why this approach was used. The authors need to include this information.

Figures showing proliferation data could include 24h old BrdU cells in the Y axis title. This will not work for survival data as cells were 32 to 56d old.

Lastly, phenotypic analysis of surviving BrdU cells (neuron vs. glia) were not performed. It may be important for the authors to discuss that while rate of survival was altered in P6/P14 the phenotype of the cells is unknown. If tissue is available this may be an important data set to add to the study.

Round 2

Reviewer 1 Report

The manuscript demonstrates, in essence, whether zinc is involved in the post-natal periods. This is partly shown in the past. See for example citation below. Moreover, in the revised manuscript, the figures say “revised to include 24h old BrdU+ cells, but the graphs look identical to the old version. And no additional work has been performed to validate their observations. I have, unfortunately, to reiterate that this paper fails to validate their observations as suggested in the previous comments. Primarily, the article lacks novelty. In the revision, letter authors mention that the study is novel and should be regarded as a starting point or a foundation for future studies. However, some reports already indicated cell proliferation and neuronal differentiation alterations in ZnT3 Ko mice. See, for example, PMC7496127. In conclusion, while I acknowledge the huge amount of work performed by the authors, I have unfortunately to reiterate that this paper falls short in validating their observations.